# Myeloid cell deficiency of the inflammatory transcription factor Stat4 protects long-term synaptic plasticity from the effects of a high-fat, high-cholesterol diet

Xiao-lei Zhang[1], Callie M. Hollander [1], Mohammad Yasir Khan[2], Melinee D'silva[2], Haoqin Ma [1], Xinyuan Yang[1], Robin Bai[3], Coles K. Keeter[3], Elena V. Galkina[3,4], Jerry L. Nadler[2,5] & Patric K. Stanton [1✉]

Neuroinflammation is associated with neurodegenerative diseases, including Alzheimer's and Parkinson's. The cytokine interleukin-12 activates signal transducer and activator of transcription 4 (Stat4), and consumption of a high-fat, high-cholesterol diet (HFD-C) and Stat4 activity are associated with inflammation, atherosclerosis, and a diabetic metabolic phenotype. In studies of in vitro hippocampal slices from control $Stat4^{fl/fl}Ldlr^{-/-}$ mice fed a HFD-C diabetogenic diet, we show that Schaffer collateral-CA1 synapses exhibited larger reductions in activity-dependent, long-term potentiation (LTP) of synaptic transmission, compared to mice fed a standard diet. Glucose tolerance and insulin sensitivity shifts produced by HFD-C diet were reduced in $Stat4^{\Delta LysM}Ldlr^{-/-}$ mice compared to $Stat4^{fl/fl}Ldlr^{-/-}$ controls. $Stat4^{\Delta LysM}Ldlr^{-/-}$ mice, which lack Stat4 under control of the $LysM^{Cre}$ promoter, were resistant to HFD-C induced impairments in LTP. In contrast, Schaffer collateral-CA1 synapses in $Stat4^{\Delta LysM}Ldlr^{-/-}$ mice fed the HFD-C diet showed *larger* LTP than control $Stat4^{fl/fl}Ldlr^{-/-}$ mice. Expression of a number of neuroinflammatory and synaptic plasticity genes was reduced by HFD-C diet in control mice, and less affected by HFD-C diet in $Stat4^{\Delta LysM}Ldlr^{-/-}$ mice. These data suggest that suppression of Stat4 activation may protect against effects of Western diet on cognition, type 2 diabetes, and reduce risk of Alzheimer's disease and other neurodegenerative disorders associated with neuroinflammation.

[1] Department of Cell Biology & Anatomy, New York Medical College, Valhalla, NY 10595, USA. [2] Department of Pharmacology, New York Medical College, Valhalla, NY 10595, USA. [3] Department of Microbiology & Molecular Cell Biology, Eastern Virginia Medical School, Norfolk, VA 23507, USA. [4] Center for Integrative Neuroscience and Inflammatory Diseases, Eastern Virginia Medical School, Norfolk, VA 23507, USA. [5] ACOS-Research VA Northern California Health Care System, Sacramento, CA 95655, USA. ✉email: patric_stanton@nymc.edu

Chronic inflammation is an important driver of many pathologies including Alzheimer's disease (AD), type 2 diabetes, and atherosclerosis[1]. As of 2020, an average of 5.8 million Americans suffer from AD, a chronic progressive inflammatory neurodegenerative disease that is the most common form of dementia in the elderly[2]. Evidence suggests that AD may begin 20 years or more before onset of symptoms[2], which could provide a wider window for preventing AD and/or halting disease progression. Conditions that promote cardiovascular disease, including type 2 diabetes, aging, hyperlipidemia, and vascular dysfunction, are associated with a higher risk of developing AD, highlighting potential common mechanisms and cross-talk between these pathologies[3–8]. While specific pathways that accelerate development of neuroinflammation and subsequently AD are not well-understood, evidence indicates that cytokines play a critical role in shaping neuroinflammation[9] and atherosclerosis development; therefore, cytokines can be important players connecting and supporting development of both pathologies.

Interleukin-12 (IL-12) is a major proinflammatory cytokine, with IL-12-dependent pathways being involved in AD-associated neuroinflammation and cognitive defects[10,11]. IL-12-related genes are linked to cognitive changes associated with aging[12] and IL-12 is a predictor of β-Amyloid load in an AD cohort[13,14]. IL-12 is upregulated in microglia of AD-like transgenic mice and targeting the IL-12p40 subunit reduced amyloid pathologic changes and ameliorated cognitive defects[14]. IL-12 leads to JAK-dependent phosphorylation and activation of signal transducer and activator of transcription 4 (Stat4), a member of the Stat family of transcription factors[15]. Stat4-deficiency leads to improved metabolic phenotype[16] and attenuated atherogenesis in several murine models of atherosclerosis[17,18]. The Human Protein Atlas reported high mRNA and protein expression of Stat4 within the brain, with the most Stat4 detected in neurons, highlighting a potential role of the IL-12/Stat4 axis directly in non-hematopoietic cells. While the role of IL-12 in AD pathology and atherosclerosis is relatively established, the role of Stat4 in neuroinflammation and associated AD structural or functional pathology under pro-atherogenic conditions has not been investigated.

Evidence suggests that AD-associated cognitive deficits result from impairments in activity-dependent long-term synaptic plasticity essential for memory consolidation. Long-term potentiation (LTP) is elicited by high-frequency bursts of synaptic stimulation that persistently increase the strength of synaptic connections between neurons[19]. LTP is impaired both in animal models of AD[20,21] and in normal aging[22], and these impairments precede the appearance of measurable impairments in learning and memory[21]. As atherosclerosis and often type 2 diabetes are associated with AD and neuroinflammation, it is possible that these pathologies might alter LTP and support development of neuroinflammation, and subsequently, AD.

Here, we utilized the low-density lipoprotein receptor-deficient ($Ldlr^{-/-}$) mouse paired with a high-fat, high-cholesterol diet (HFD-C) as an established model of hypercholesterolemia, atherosclerosis, metabolic dysfunction, and systemic inflammation, and investigated whether these conditions might promote defective LTP. We further characterized the role of Stat4 in shaping these defects, using HFD-C fed $Ldlr^{-/-}$ mice with a LysM$^{cre}$-dependent deficiency of Stat4 ($Stat4^{ΔLysM}Ldlr$). Stat4 deficiency, under control of LysM$^{Cre}$ promoter activity, improved whole-body metabolic phenotype and rescued deficits in LTP elicited by 16 weeks feeding with the HFD-C diet in $Stat4^{ΔLysM}Ldlr^{-/-}$ mice.

## Results

### $Stat4^{ΔLysM}Ldlr^{-/-}$ mice show reduced p-Stat4 expression in hippocampus. 

Vascular dysfunction caused by atherosclerosis and hyperlipidemia results in blood brain barrier breakdown,

inflammation, impaired clearance of Aβ plaques, and neurovascular dysfunction[23]. As IL-12 is involved in inflammatory processes in atherosclerosis and neuroinflammation, and Stat4 is an important component of the IL-12-dependent response[24–26], we sought to test whether Stat4 is expressed in the brain of HFD-C fed $Stat4^{fl/fl}Ldlr^{-/-}$ atherosclerotic mice. $Ldlr^{-/-}$ mice is a well-established model of atherosclerosis and is often used to investigate pathology of atherosclerosis. HFD-C feeding results in an elevation of total circulating cholesterol (up to ~1000 mg/dl) and the LysMCre promoter does not modify these changes.

As shown in Fig. 1, we detected Stat4 expression within the brain of HFD-C fed $Stat4^{fl/fl}Ldlr^{-/-}$ mice. To gain further insights into the potential role of Stat4 in the regulation of neuroinflammation, we used a newly generated $Stat4^{ΔLysM}Ldlr^{-/-}$ mice and tested expression of pStat4 in the brain. As the $Stat4^{fl/fl}$ mice is a relatively new strain, we used $Stat4^{fl/fl}Ldlr^{-/-}$ mice as controls, to

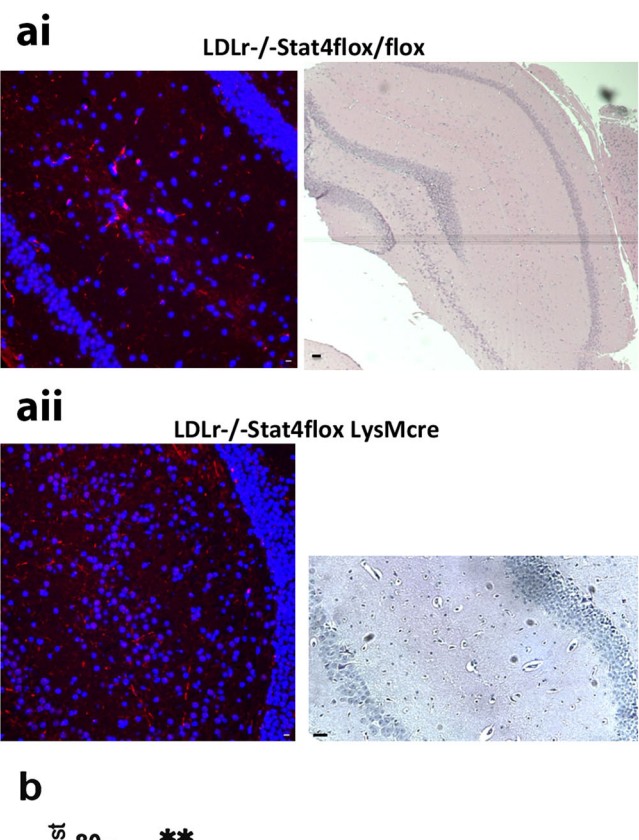

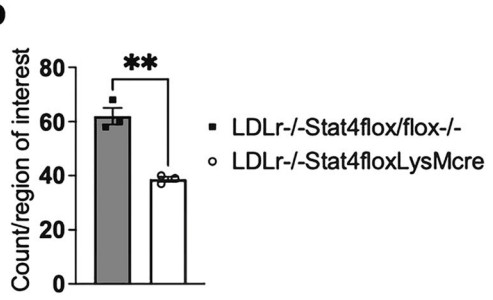

**Fig. 1 $Stat4^{fl/f}LysM^{Cre}Ldlr^{-/-}$ mice show a partial lack of expression of pStat4. a** Representative image of Immunofluorescence staining of pStat4 (red) and nuclei (blue) (**ai**) and hematoxylin/eosin stained sequential section (**aii**) in the dentate gyrus of the hippocampal formation, in a control $Stat4^{fl/fl}Ldlr^{-/-}$ versus a $Stat4^{fl/fl}LysM^{Cre}Ldlr^{-/-}$ mice fed the HFD-C diet for 16 weeks. **b** Mean ± SEM of pStat4 immunofluorescence staining ($n = 4$ biologically independent animals per genotype), showing that $Stat4^{fl/fl}LysM^{Cre}Ldlr^{-/-}$ mice exhibited significantly reduced pStat4 labeling in the hippocampus compared to $Stat4^{fl/fl}Ldlr^{-/-}$ mice, consistent with a partial lack of Stat4 expression and activation. **$p = 0.0018$, scale bars 20 μm.

account for potential effects of $Stat4^{fl/fl}$ on synaptic transmission and plasticity. LysM, that encodes lysozyme in mice, is widely used to modulate gene expression in myeloid cells[27,28]. Thus, LysM-Cre can be used to investigate the role of myeloid cells in CNS activation[29]. Importantly, an increasing body of evidence also suggests that the LysM promoter is also expressed in neurons in the brain[30].

Figure 1 illustrates the reduction in phosphorylation of Stat4 observed in $Stat4^{\Delta LysM}Ldlr^{-/-}$ mice fed the HFD-C diet, compared with control $Stat4^{fl/fl}Ldlr^{-/-}$ mice. As summarized in Fig. 1b, knockout of Stat4 expression using this LysM$^{cre}$ promoter produced an approximately 50% reduction in overall Stat4+ cell labeling, in mice fed the HFD-C diet from 8–24 weeks of age. This confirms that cells in this region of the hippocampus express activated Stat4, and that there are some residual cell types still expressing Stat4 in the $Stat4^{\Delta LysM}Ldlr^{-/-}$ mice.

**Lack of Stat4 in $Stat4^{\Delta LysM}Ldlr^{-/-}$ mice protects against metabolic effects of a high-fat diet.** HFD-C diet has been shown previously to induce glucose intolerance and insulin resistance in 16 wks HFD-C fed $Ldlr^{-/-}$ mice[31]. To test to what extent Stat4-deficiency in myeloid cells improved metabolic phenotype, we performed glucose tolerance and insulin tolerance tests on $Stat4^{\Delta LysM}Ldlr^{-/-}$ and $Stat4^{fl/fl}Ldlr^{-/-}$ mice. As shown in Fig. 2, 16 week HFD-C fed $Stat4^{\Delta LysM}Ldlr^{-/-}$ mice showed both significantly improved blood glucose control in response to a glucose challenge (Fig. 2a), and improved insulin sensitivity (Fig. 2b). These findings point to a potential role for STAT4 under activity of the LysM$^{Cre}$ promoter in regulating the disturbed metabolic phenotype triggered by the HFD-C diet.

**Lack of Stat4 reduces expression of inflammatory pathway genes in brain.** To characterize the effects of a lack of Stat4 on the expression patterns of inflammatory pathway genes, we utilized Nanostring™ technology to compare expression patterns of 758 genes across 50+ pathways in the hippocampus of $Stat4^{\Delta LysM}Ldlr^{-/-}$ mice versus $Stat4^{fl/fl}Ldlr^{-/-}$ control mice fed a normal chow diet to 24 weeks of age. Figure 3 shows the collection of genes that were significantly up or down-regulated in expression in brains of chow–diet fed $Stat4^{\Delta LysM}Ldlr^{-/-}$ versus $Stat4^{fl/fl}Ldlr^{-/-}$ mice. One gene down-regulated in $Stat4^{\Delta LysM}Ldlr^{-/-}$ brain was *Tirap*, a toll-interleukin 1 receptor domain-containing adaptor protein, that mediates protein-protein interactions between the toll-like receptors and signal-transduction components. Interestingly, *Mapk38* expression was also down-regulated in $Stat4^{\Delta LysM}Ldlr^{-/-}$ mice, suggesting that Stat4-deficiency might

also down-regulate cell proliferation and differentiation, as well as the expression of key inflammatory mediators, including cytokines. Indeed, a large portion of genes that were down-regulated were associated with interleukin immune system modulation of inflammation (*Il12a, Il2r, Il17rb, il21r, Myd88*, caspase 1/interleukin-1 converting enzyme), consistent with the requirement for Stat4 in IL-12 mediated immune responses.

One of the most upregulated genes was *Fpr1*, a gene that is involved in a response of phagocytic cells to invasion of the host by microorganisms. Interestingly, the three genes that were significantly up-regulated are all associated with antiviral responses (*Infz, Infa, Ifitm1*, encoding interferon induced transmembrane protein 1). Lack of Stat4 also down-regulated expression of *Itpr3* encoding inositol 1,4,5-triphosphate receptor 3, which triggers release of $Ca^{2+}$ from intracellular stores, and two protein tyrosine phosphatases (non-receptor type 6, receptor type C) expressed largely in haematopoietic cells, which regulate T and B cell antigen receptor-mediated activation. Additional WikiPathway analysis further highlighted Interleukins IL-1, IL-2, IL-3, IL-5, and IL-6, and MAPK-signaling pathways associated with the regulation of inflammation (Fig. 3c). Interestingly, fibrin complement receptor 3 pathway genes were also significantly different between $Stat4^{\Delta LysM}Ldlr^{-/-}$ and $Stat4^{fl/fl}Ldlr^{-/-}$ mice. As fibrin activates microglia, resulting in PI3K/NF-kB activation and production of inflammatory cytokines, our data point to a potential role for Stat4 in microglia activation. It is worth noting that apoptosis pathway genes were also altered between chow fed $Stat4^{\Delta LysM}Ldlr^{-/-}$ and $Stat4^{fl/fl}Ldlr^{-/-}$ mice (Fig. 3c), further suggesting a role for Stat4 in regulating susceptibility of CNS cells to programmed cell death.

**Lack of Stat4 reduces activation by HFD-C diet of genes associated with inflammation, synaptic plasticity and AD.** Sixteen weeks feeding with the HFD-C diet also resulted in differential gene expression patterns between $Stat4^{\Delta LysM}Ldlr^{-/-}$ and $Stat4^{fl/fl}Ldlr^{-/-}$ control mice. Figure 4 shows the collection of genes that were significantly up or down-regulated in expression in brains of $Stat4^{\Delta LysM}Ldlr^{-/-}$ compared to $Stat4^{fl/fl}Ldlr^{-/-}$ mice, when both were fed the HFD-C diet for 16 weeks. We found 77 differentially regulated genes between HFD-C fed $Stat4^{\Delta LysM}Ldlr^{-/-}$ and $Stat4^{flox/flox}Ldlr^{-/-}$ brain tissues (Fig. 4c). A few genes that were upregulated within the brain of HFD-C fed $Stat4^{\Delta LysM}Ldlr^{-/-}$ mice were genes involved in leukocyte migration, including, *Ccrl1, ITtgb7*, encoding chemokine (C-C motif) receptor 1-like 1 and integrin beta 7, respectively, and *Cd6* gene that encodes protein with a binding site for an activated leukocyte cell adhesion molecule.

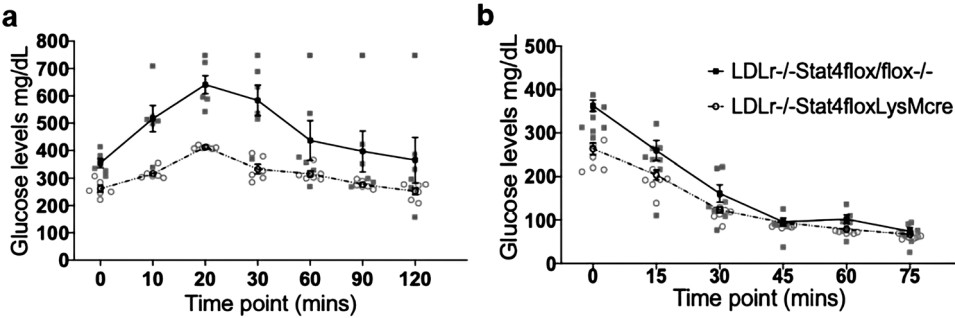

**Fig. 2 $Stat4^{fl/f}LysM^{Cre}Ldlr^{-/-}$ mice exhibit improved glucose tolerance and response to insulin when challenged by feeding for 16 weeks with a HFD-C diet. a** Glucose tolerance test (GTT) and (**b**) insulin tolerance tests (ITT) at different time points from $Stat4^{fl/f}LysM^{Cre}Ldlr^{-/-}$ and $Stat4^{fl/fl}Ldlr^{-/-}$ mice on HFD-C diet for 16 weeks starting at 8 weeks old ($n = 8$ biologically independent animals of each sex per genotype). Two-way ANOVA for repeated measures shows that GTT (genotype factor $p = 0.0068$, df = 10) and ITT (genotype factor $p = 0.0118$, df = 12) both exhibited significantly improved responses to glucose and insulin, respectively, in animals lacking full expression of Stat4, after mice were challenged by feeding of the HFD-C diet for 24 weeks. All error bars are standard error of the mean.

**a**

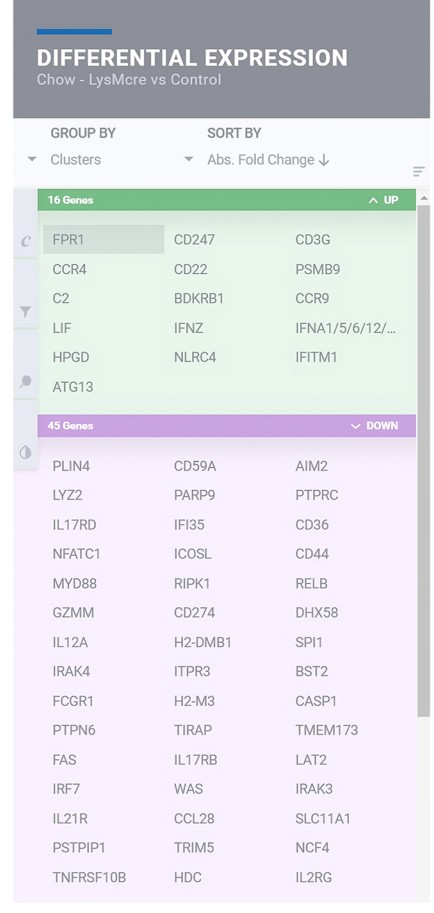

**b**

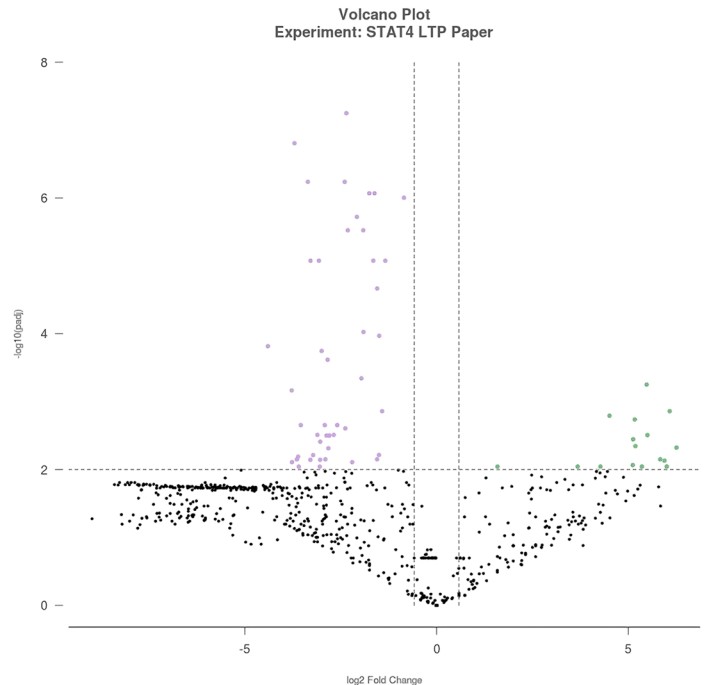

**c**

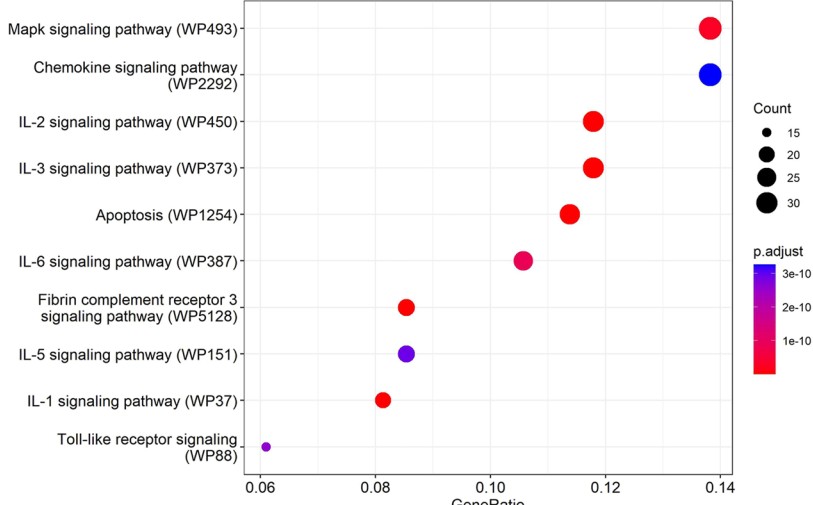

A number of genes that were significantly down-regulated in $Stat4^{\Delta LysM}Ldlr^{-/-}$ mice challenged by the HFD-C diet were, similar to chow diet, associated with interleukin-dependent immune system modulation. WikiPathway analysis further highlighted the chemokine- and IL-2 and IL-3-dependent pathways that are altered in $Stat4^{\Delta LysM}Ldlr^{-/-}$ mice (Fig. 4c).

Interestingly, IL-3 has recently been reported to be involved in AD suppression via microglial activation[32]. *Csf1* gene, encoding colony stimulating factor 1, which controls development, maintenance, activation and survival of microglia associated with neuroinflammation, was also reduced in $Stat4^{\Delta LysM}Ldlr^{-/-}$ mice.

**Fig. 3 Nanostring™ inflammation gene expression patterns in $Stat4^{fl/fl}LysM^{Cre}Ldlr^{-/-}$ mice lacking Stat4, compared to control $Stat4^{fl/fl}Ldlr^{-/-}$ mice, when fed a normal chow diet. a–b** Table and volcano plot showing significantly up (green) and down (purple) regulated genes in the hippocampus $Stat4^{fl/fl}LysM^{Cre}Ldlr^{-/-}$ versus age-matched control $Stat4^{fl/fl}Ldlr^{-/-}$ mice fed a chow diet at 24 weeks of age. Differentially expressed genes were determined as >1.5-fold difference with adjusted $p$ value < 0.05. $n = 3$ biologically independent animals per group. **c** Dot plot shows the altered GO terms (FDR < 0.05) of biological processes and molecular functions identified using DAVID to be enriched under the different contrasts. Dot plot shows the up-regulated KEGG pathways (FDR < 0.1) enriched for different contrasts. The size of the dot is based on gene count enriched in the pathway, and the color of the dot shows the pathway enrichment significance. Dot plot shows the down-regulated GO terms (FDR < 0.05) of biological processes and molecular functions identified using WikiPathway analysis. The size of the dot is based on gene count enriched in the pathway, and the color of the dot shows the pathway enrichment significance.

Thioredoxin-interacting protein (TXNIP) also exhibited reduced expression in $Stat4^{\Delta LysM}Ldlr^{-/-}$ mice challenged by the HFD-C diet, compared to $Stat4^{fl/fl}Ldlr^{-/-}$ control mice. TXNIP interacts with and regulates thioredoxin. Over the past few years, it has become clear that increases in TXNIP leads to insulin resistance and reduced pancreatic beta cell production of insulin[33]. In contrast, animals with TXNIP deficiency are protected from diet-induced insulin resistance and diabetes[34]. The reduction of TXNIP in the brains of the HFD-C mice with LysM$^{cre}$-driven deletion of Stat4 suggests that targeted Stat4 deletion could maintain insulin signaling despite the high fat diet.

Other down-regulated genes are associated with synaptic transmission and plasticity: *Adora2ap* (encoding adenosine A2A receptor), *Mapkapk2* (encoding map kinase-activated protein kinase 2), *Rab7*, RAS guanyl releasing protein 1 (*Rasgrp1*). Adenosine A2A receptors, via production of cyclic AMP, enhance release of neurotransmitters, including glutamate, acetylcholine and dopamine, which promotes the induction of long-term potentiation (LTP) of synaptic transmission. MAPKAP2, Rab7 and RASGRP1 are members of the Ras oncogenic pathway, which activate Erk/MAPkinase pathways that lead to LTP. In line with these gene expression changes, WikiPathway analysis also showed alterations in MAPK-and toll-like receptor-dependent pathways for $Stat4^{\Delta LysM}Ldlr^{-/-}$ compared with $Stat4^{fl/fl}Ldlr^{-/-}$ mice (Fig. 4c). One fascinating gene with reduced expression levels in $Stat4^{\Delta LysM}Ldlr^{-/-}$, compared with $Stat4^{fl/fl}Ldlr^{-/-}$ mice, when challenged with the HFD-C diet, was amyloid beta precursor protein (APP), supporting our as yet untested hypothesis that Stat4 plays a role, through neuroinflammation, in the progression of AD.

**Lack of Stat4 expressed under the $LysM^{Cre}$ promoter does not alter the magnitude of LTP at 3 weeks of age.** Figure 5a shows the time course, and Fig. 5b mean ± SEM magnitude, of the expression of LTP at Schaffer collateral-CA1 synapses in hippocampal slices from 3 week old $Stat4^{\Delta LysM}Ldlr^{-/-}$ (open circles) versus $Stat4^{fl/fl}Ldlr^{-/-}$ control mice (filled circles), following the application of 3 theta burst trains of high frequency stimulation (arrow, TBS), in n slices per group. In slices from three week old mice nursed by dams fed a normal chow diet, the magnitude of LTP at 60 min post-TBS was not significantly different in $Stat4^{\Delta LysM}Ldlr^{-/-}$ mice, compared to age-matched control $Stat4^{fl/fl}Ldlr^{-/-}$ mice with normal Stat4 expression. At this early developmental stage, a lack of Stat4 did not impair the expression of long-term, activity-dependent synaptic potentiation at Schaffer collateral-CA1 synapses in the hippocampus.

**Consumption of a high-fat/high-cholesterol diet for 16 weeks during early adulthood markedly impairs LTP.** In contrast to young 3 week old mice nursed by dams fed a chow diet, when $Stat4^{\Delta LysM}Ldlr^{-/-}$ and $Stat4^{fl/fl}Ldlr^{-/-}$ mice were challenged by prolonged feeding on the HFD-C, the responses between the two groups were strikingly different, as illustrated in Fig. 6. Chow fed $Stat4^{fl/fl}Ldlr^{-/-}$ mice at 24 weeks of age exhibited robust LTP of

1.76 ± 0.14 times pre-TBS baseline responses at 60 min post-TBS. When $Stat4^{fl/fl}Ldlr^{-/-}$ mice were fed the HFD-C diet from 8–24 weeks of age, the magnitude of LTP was statistically significantly reduced 60 min post-TBS to 1.25 ± 0.04 times pre-TBS baseline, indicating that the HFD-C diet impairs the long-term, activity-dependent synaptic plasticity necessary for normal cognition, learning and memory (Fig. 6).

**Lack of Stat4 expressed under the $LysM^{Cre}$ promoter rescues effects of the HFD-C diet on LTP.** In contrast to the effects of the HFD-C diet in $Stat4^{fl/fl}Ldlr^{-/-}$ control mice, $Stat4^{\Delta LysM}Ldlr^{-/-}$ mice were resistant to HFD-C induced impairments in expression of LTP (Fig. 7, $Stat4^{\Delta LysM}Ldlr^{-/-}$, filled circles and black bars). LTP in the absence of Stat4 was not significantly different in slices from $Stat4^{\Delta LysM}Ldlr^{-/-}$ mice fed normal chow, compared to these mice fed HFD-C diets. These data demonstrate that Stat4 is a key contributor to the chronic effects of the HFD-C diet that results in impairments in LTP, and supports the hypothesis that this inflammatory transcription factor is a key mediator of neuroinflammation-induced impairments in long-term activity-dependent synaptic plasticity. Previous studies have shown that Stat4 plays a critical role in HFD-C diet induced cardiovascular inflammation[17,18], supporting the hypothesis that Stat4 also acts via vascular inflammation in the brain to accelerate HFD-C induced impairments in synaptic plasticity that underlie associated cognitive impairments, and may play an important role in the heightened risk of AD in type 2 diabetes.

**Lack of Stat4 expressed under the LysM$^{Cre}$ promoter does not alter basal synaptic transmission or paired-pulse facilitation at 3 or 24 weeks of age.** The above results indicate that prolonged consumption of the HFD-C diet impairs LTP in mice with normal expression of Stat4. To determine whether the decrease in LTP was a result of alterations in basal synaptic transmission, we compared paired-pulse facilitation, and basal input-output relations, at Schaffer collateral-CA1 synapses in hippocampal slices from $Stat4^{\Delta LysM}Ldlr^{-/-}$ and $Stat4^{fl/fl}Ldlr^{-/-}$ mice after 16 weeks feeding with the HFD-C diet.

Figure 8a shows paired-pulse profiles of the ratios of second to first EPSP slope magnitude as a function of inter-stimulus interval between the first and second stimulus, in milliseconds, in slices from 24 week old mice fed the HFD-C diet from weeks 8–24 (N = 11 Control $Stat4^{fl/fl}Ldlr^{-/-}$, 11 $Stat4^{\Delta LysM}Ldlr^{-/-}$ slices). One-way ANOVA with repeated measures on stimulus intervals showed no significant difference between PPFs evoked in control $Stat4^{fl/fl}Ldlr^{-/-}$ and $Stat4^{\Delta LysM}Ldlr^{-/-}$ slices. Thus, the decrease in magnitude of LTP elicited by the HFD-C diet in $Stat4^{fl/fl}Ldlr^{-/-}$ mice, compared to mice fed a normal chow diet lacking Stat4 ($Stat4^{\Delta LysM}Ldlr^{-/-}$), was not a result of alterations in presynaptic transmitter release probability as assessed by paired-pulse facilitation profiles.

Figure 8b shows profile of PPFs recorded from $Stat4^{fl/fl}Ldlr^{-/-}$ and $Stat4^{\Delta LysM}Ldlr^{-/-}$ groups slices at 3 weeks of age (N = 12 in

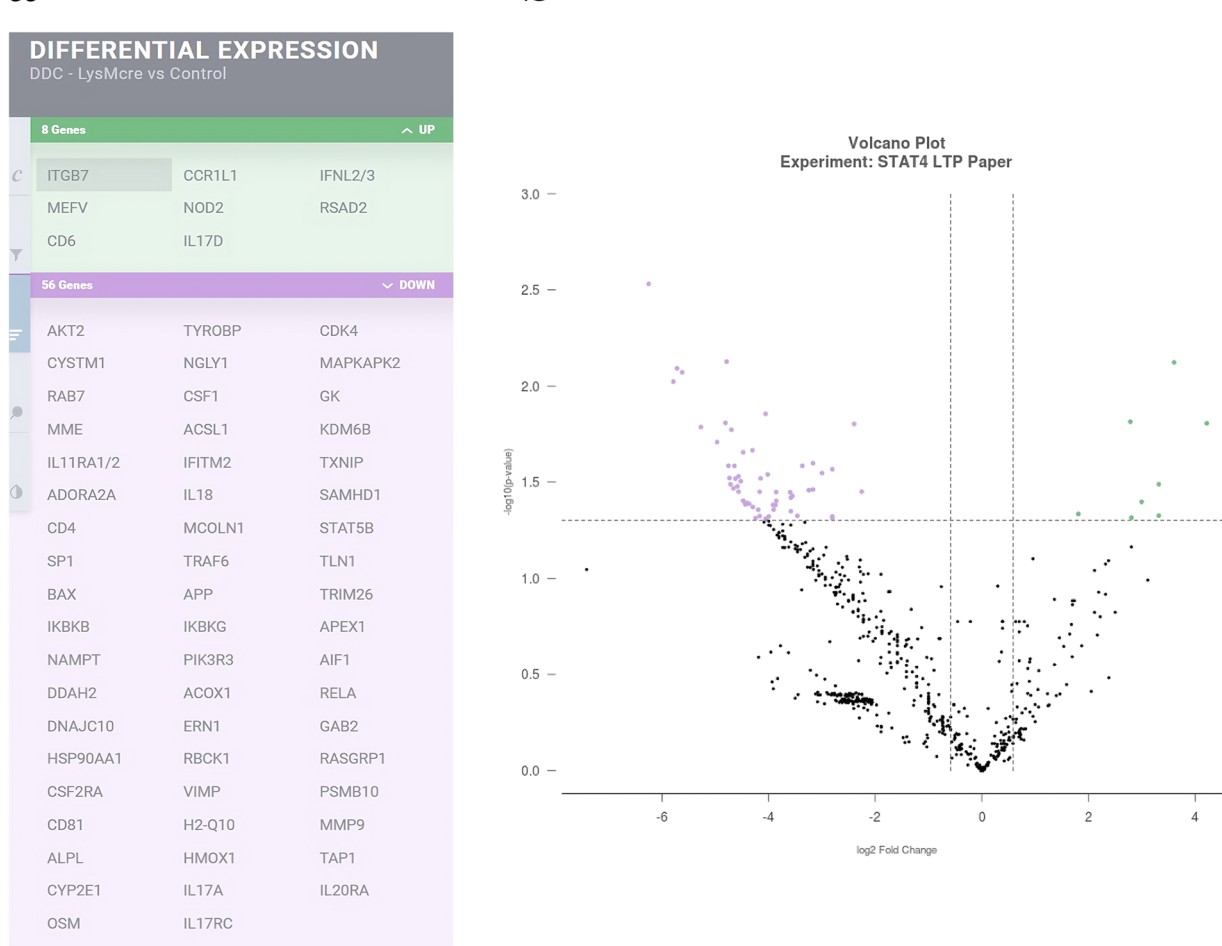

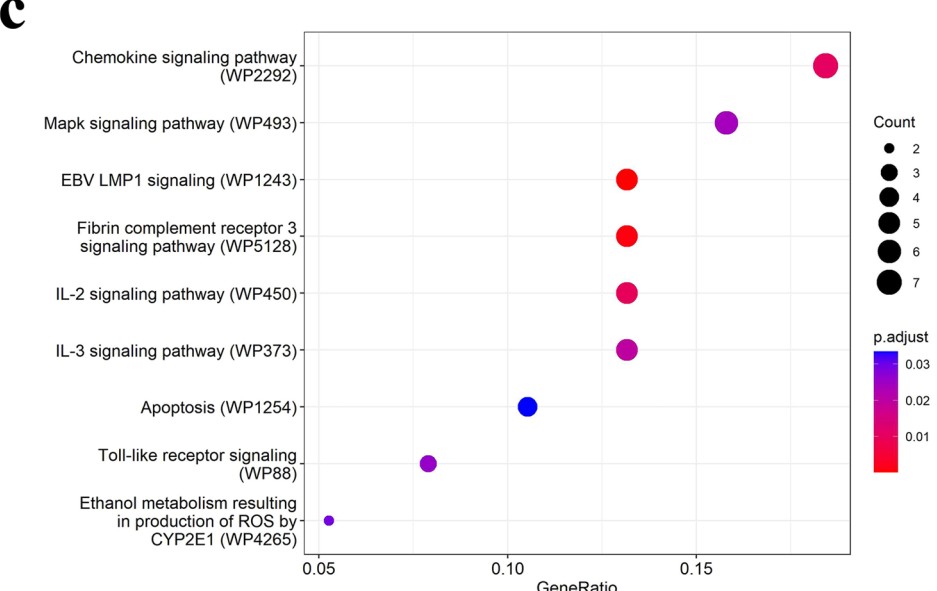

each group). One-way ANOVA with repeated measures on stimulus intervals showed no significance differences between groups, indicating that elimination of Stat4 under LysM$^{cre}$ promoter activity did not alter PPF in 3-week-old mice, suggesting normal early development of presynaptic vesicular release in neurons lacking Stat4.

Figure 9 shows the input/output relationships of normalized EPSP amplitude (y axis) to stimulus intensity (x axis) at 24 weeks of age on HFD-C diet (Fig. 9a, $N = 15$ control $Stat4^{fl/fl}Ldlr^{-/-}$ slices, $N = 15$ $Stat4^{\Delta LysM}Ldlr^{-/-}$ slices), and at 3 weeks of age (Fig. 9b, $N = 21$: $Stat4^{fl/fl}Ldlr^{-/-}$ slices, $N = 21$ $Stat4^{\Delta LysM}Ldlr^{-/-}$ slices). These synaptic strength profiles exhibited no differences in

**Fig. 4 Nanostring™ inflammation gene expression patterns in *Stat4$^{fl/f}$LysM$^{Cre}$Ldlr$^{−/−}$* mice lacking Stat4, compared to control *Stat4$^{fl/fl}$ Ldlr$^{−/−}$* mice, when fed a HFD-C diet for 16 weeks. a–b** Volcano plot and table showing significantly up (green) and down (purple) regulated genes in the hippocampus of *Stat4$^{fl/f}$LysM$^{Cre}$Ldlr$^{−/−}$* versus age-matched control *Stat4$^{fl/fl}$ Ldlr$^{−/−}$* mice fed the HFD-C diet for 16 weeks (from 8–24 weeks of age). Differentially expressed genes were determined as >1.5-fold difference with adjusted *p* value < 0.05. *n* = 3 biologically independent animals per group. **c** Dot plot shows the altered GO terms (FDR < 0.05) of biological processes and molecular functions identified using DAVID to be enriched under the different contrasts. Dot plot shows the up-regulated KEGG pathways (FDR < 0.1) enriched for different contrasts. The size of the dot is based on gene count enriched in the pathway, and the color of the dot shows the pathway enrichment significance. Dot plot shows the down-regulated GO terms (FDR < 0.05) of biological processes and molecular functions identified using WikiPathway analysis. The size of the dot is based on gene count enriched in the pathway, and the color of the dot shows the pathway enrichment significance.

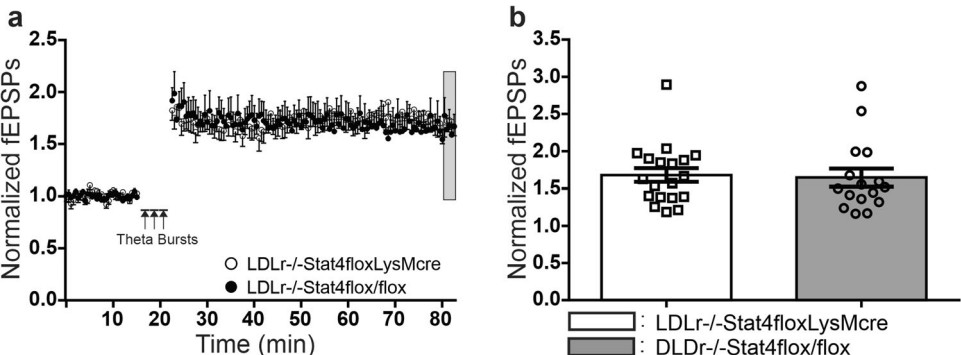

**Fig. 5 *Stat4$^{fl/f}$LysM$^{Cre}$Ldlr$^{−/−}$* versus age-matched control *Stat4$^{fl/fl}$ Ldlr$^{−/−}$* mice at 3 weeks of age exhibit similar magnitude LTP at Schaffer collateral-CA1 synapses in hippocampus. a** Time course of LTP following three theta burst trains of stimulation (TBS ×3) of Schaffer collateral axons spaced 3 min apart in hippocampal slices from *Stat4$^{fl/f}$LysM$^{Cre}$Ldlr$^{−/−}$* mice (open circles, each point mean ± SEM) versus control *Stat4$^{fl/fl}$ Ldlr$^{−/−}$* mice (filled circles, each point mean ± SEM). **b** Mean ± SEM magnitude of LTP 60 min post-TBS, showing that *Stat4$^{fl/f}$LysM$^{Cre}$Ldlr$^{−/−}$* slices (filled bars, *n* = 19 biologically independent slices, normalized LTP = 1.68 ± 0.092) exhibited unchanged LTP compared to control *Stat4$^{fl/fl}$ Ldlr$^{−/−}$* slices (open bars, *n* = 16 biologically independent slices, normalized LTP = 1.65 ± 0.121, Student's *t*-test, *p* = 0.8254, *t* = 0.2224, df = 33).

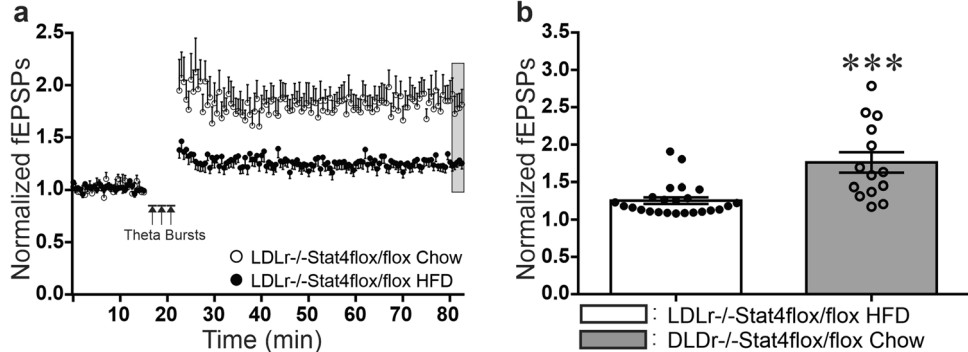

**Fig. 6 *Stat4$^{fl/f}$Ldlr$^{−/−}$* mice maintained on the HFD-C diet for 16 weeks (from 8–24 weeks of age) showed significant impairments in LTP at Schaffer collateral-CA1 synapses, compared to these control *Stat4$^{fl/f}$Ldlr$^{−/−}$* mice fed a normal chow diet. a** Time course of LTP following three theta burst trains of stimulation (TBS ×3) of Schaffer collateral axons spaced 3 min apart in hippocampal slices from *Stat4$^{fl/f}$Ldlr$^{−/−}$* mice were fed either the HFD-C diet (filled circles, *n* = 24 biologically independent slices) or chow diet (open circles, *n* = 14 biologically independent slices) for 16 weeks (each point mean ± SEM). **b** Mean ± SEM magnitude of LTP from grey region in (**a**), showing that long-term feeding with the HFD-C diet (filled bar) caused significant impairments in the magnitude of LTP, compared to chow-fed control *Stat4$^{fl/f}$Ldlr$^{−/−}$* mice (open bar). (Student's *t*-test, ***p* = 0.001, *t* = 4.286, df = 36).

basal synaptic transmission in either young mice, or mice in either group fed the HFD-C diet (One-way ANOVA) with repeated measures on stimulus intensity ($F_{(1,40)}$ = 0.674, *p* = 0.416 for 3 week-old groups and $F_{(1,28)}$ = 0.491, *p* = 0.489 for 24 week old groups). These results indicate that baseline synaptic transmission was not altered by Stat4 deficiency at either 3 weeks of age, or after feeding with the HFD-C diet from 8–24 weeks of age, in either control *Stat4$^{fl/fl}$Ldlr$^{−/−}$* or *Stat4$^{ΔLysM}$Ldlr$^{−/−}$* mice.

## Discussion

In the present study, we found that: (1) The inflammatory transcription factor Stat4 is present and activated in the hippocampus, (2) feeding control *Stat4$^{fl/fl}$Ldlr$^{−/−}$* mice a HFD-C diet for 16 weeks, from 8–24 weeks of age (20–34 years of age human equivalent), reduced long-term, activity-dependent synaptic potentiation (LTP), compared to mice of the same age fed a chow diet. (3) Stat4 deficiency under control of the LysM$^{Cre}$ promoter protected against HFD-C induced metabolic alterations, (4) Deficiency of Stat4 also completely prevented the decline in long-term, activity-dependent synaptic potentiation (LTP) produced by feeding a HFD-C diet to mice. (5) Neither the HFD-C diet, nor absence of Stat4 in myeloid cells, altered basal synaptic transmission, or presynaptic release probability assessed by paired-pulse stimulation.

Our electrophysiological studies have implications for understanding HFD-C, aging and neurodegenerative disease-related

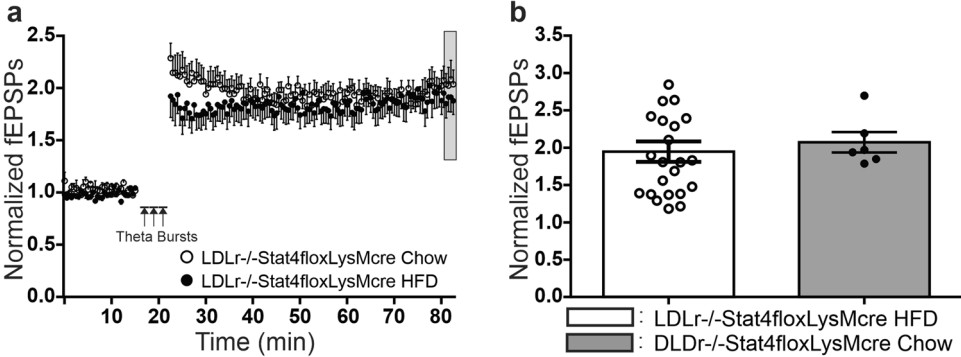

**Fig. 7 $Stat4^{fl/f}LysM^{Cre}Ldlr^{-/-}$ mice maintained either on the HFD-C diet for 16 weeks, or on chow diet for 16 weeks, show no significant differences in LTP at Schaffer collateral-CA1 synapses. a** Time course of LTP following three theta burst trains of stimulation (TBS ×3) of Schaffer collateral axons spaced 3 min apart in hippocampal slices from $Stat4^{fl/f}LysM^{Cre}Ldlr^{-/-}$ mice fed the HFD-C diet (filled circles, $n = 23$ biologically independent slices) or chow diet (open circles, $n = 6$ biologically independent slices) for 16 weeks (each point mean ± SEM). **b** Mean ± SEM LTP from grey region in (**a**), showing that a lack of Stat4 expression in myeloid cells and neurons protected LTP from the effects of prolonged feeding with the HFD-C diet (Student's two-tailed $t$-test, $t = 0.4434$, df = 27, $p = 0.661$).

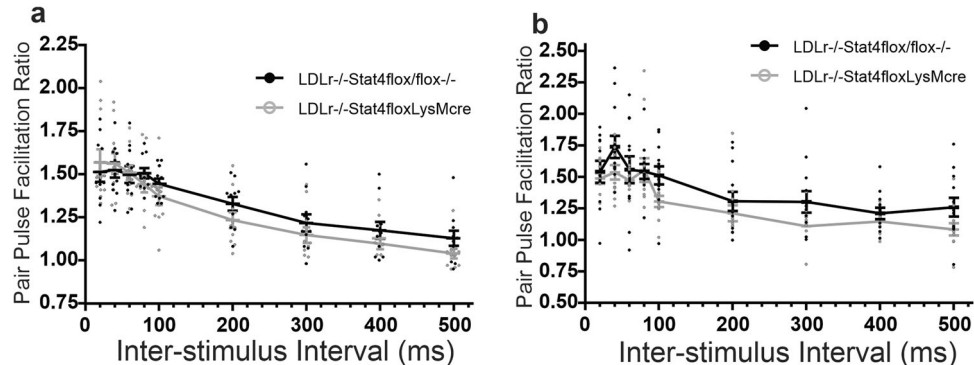

**Fig. 8 A lack of Stat4 expression in myeloid cells and neurons had no effect on paired-pulse facilitation (PPF) at Schaffer collateral-CA1 synapses in slices from $Stat4^{fl/f}LysM^{Cre}Ldlr^{-/-}$ mice (open circles), compared to $Stat4^{fl/f}Ldlr^{-/-}$ mice (filled circles). a** PPF profiles for $Stat4^{fl/f}LysM^{Cre}Ldlr^{-/-}$ mice (Ldlr−/−Stat4floxLysMCre, open circles, $N = 11$ biologically independent slices) vs. $Stat4^{fl/f}Ldlr^{-/-}$ control mice (Ldlr−/−Stat4flox/flox, filled circles, n = 11 biologically independent slices) fed the HFD-C diet for 16 weeks starting at 8 weeks of age. One-way ANOVA with repeated measures showed no significant difference between the two groups of slices (F(1,20) = 0.76, $p = 0.394$). **b** PPF profiles in $Stat4^{fl/f}LysM^{Cre}Ldlr^{-/-}$ mice (Ldlr−/−Stat4floxLysMCre, open circles, $N = 12$ biologically independent slices) at 3 weeks of age plotted against age matched $Stat4^{fl/f}Ldlr^{-/-}$ mice (Ldlr−/−Stat4flox/flox, filled circles, n = 12 biologically independent slices), nursed by a dam fed normal chow diet. One-way ANOVA for repeated measures showed no significant effect of diet F(1,22) = 2.927; $p = 0.101$. All error bars are Standard Error of the Mean.

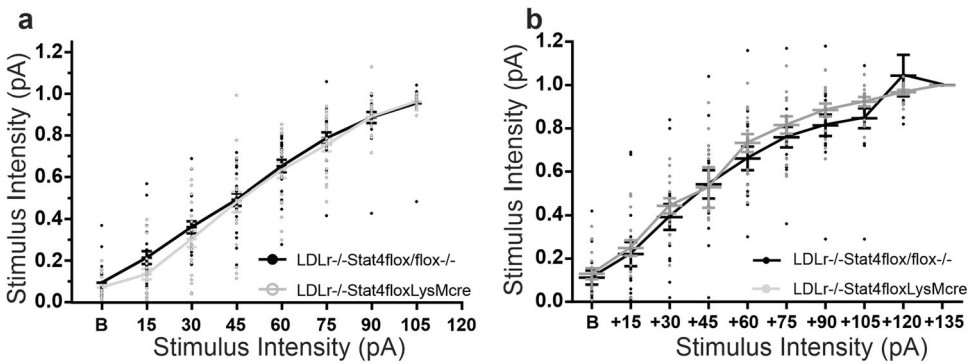

**Fig. 9 $Stat4^{fl/f}LysM^{Cre}Ldlr^{-/-}$ mice show no differences in baseline synaptic transmission at Schaffer collateral-CA1 synapses at either 3 weeks of age on normal diet, or after 16 weeks on the HFD-C diet, compared to age-matched $Stat4^{fl/f}Ldlr^{-/-}$ mice.** Mean ± SEM normalized Schaffer collateral-CA1 fEPSP slope versus stimulus intensity in the same groups of hippocampal slices where PPF was measured in Fig. 8, from 24 week old (**a**) and 3 week old (**b**) $Stat4^{fl/f}Ldlr^{-/-}$ (Ldlr−/−Stat4flox/flox, filled circles) versus $Stat4^{fl/f}LysM^{Cre}Ldlr^{-/-}$ mice at 3 weeks of age fed chow, or after 16 weeks on the HFD-C diet (Ldlr−/−Stat4floxLysMCre, open circles). None of these groups showed a significant effect of diet (One-way ANOVA for repeated measures).

declines in cognitive function. The HFD-C induced impairments in long-term plasticity we observed in adult mice is consistent with reports that such diets can alter the developmental time course of activity-dependent synaptic plasticity in the non-diseased brain[35,36], and induce cognitive impairments[37–39]. These observations support the hypothesis that activation of inflammatory mechanisms triggered by such diets are potentially an important factor in the decline in cognitive function that occurs in normal aging.

Characterization of gene expression patterns comparing mice lacking Stat4 under control of the $LysM^{cre}$ promoter, to $Stat4^{fl/fl}Ldlr^{-/-}$ control mice, revealed widespread down-regulation of inflammatory pathway genes when mice were fed either normal chow or HFD-C diets. The majority of these genes were components of interleukin and interferon signaling pathways, consistent with previous reports that serine and tyrosine phosphorylation of Stat4 is necessary for IL-12 mediated secretion of IFNγ. Stat4 deficiency also affected MAPK-signaling pathway for both chow and HFD-C regimen, highlighting a role of Stat4 in the key biological processes of proliferation, migration, and regulation of survival and death. The HFD-C diet resulted in significantly smaller increases in expression of interleukin and interferon pathway genes in $Stat4^{\Delta LysM}Ldlr^{-/-}$ mice, compared to controls fed the same diet. These data further support the hypothesis that inflammatory pathways activated by Stat4 are important contributors to HFD-C induced impairments in synaptic plasticity and cognitive function, and that reducing Stat4 expression in LysM$^{cre}$-dependent cells has neuroprotective anti-neuroinflammatory actions when challenged with the HFD-C diet. The fact that mice lacking Stat4 under LysM$^{cre}$ promoter control are resistant to HFD-C induced inflammation and impairments in synaptic plasticity suggest an important role for inflammatory pathways regulated by Stat4 in mediating impairments in synaptic plasticity, and possibly cognitive function, associated with normal aging.

$Stat4^{\Delta LysM}Ldlr^{-/-}$ mice also exhibit very intriguing differences in the response of some neuronal genes associated with synaptic plasticity to consumption of the HFD-C diet. In particular, A2A adenosine receptors, which promote release of multiple neurotransmitters including glutamate, can promote the induction of LTP of synaptic strength, while MPKP1, Rab7 and RASGRP1 are all members of Ras oncogene pathways that can activate Erk/MAPKinase necessary for some forms of LTP. However, perhaps the most intriguing observation was that the lack of Stat4 protected against the up-regulation of expression of amyloid beta precursor protein produced by the HFD-C diet. Since recent studies suggest that a combination of beta amyloid and neuroinflammation is closely associated with AD progression, Stat4 may be a prime target for suppressing both of these HFD-C associated factors, and thus for treatment of AD. Since Stat4 activation plays a key role in atherosclerosis[17,18], vascular inflammation via Stat4 activation in neurovascular compartments may also contribute to elevating risk for AD in type-2 diabetes[3–8].

The extensive literature implicating inflammation in diabetes and many progressive neurodegenerative diseases, including AD, makes Stat4 a therapeutic target with immense potential for slowing, or even preventing, neurodegenerative disease progression. We hypothesize that Stat4 may be a final common pathway mediator of inflammatory damage that is important in normal aging, diabetes and AD, and that therapies that prevent Stat4 activation have potential as novel treatments for such slow, progressive neurodegenerative diseases.

There are clear interconnections between inflammation, metabolic and vascular disease, and increased risk of AD. Neuroinflammation and associated cellular immune activation play a central role in the pathogenesis of AD[40–43]. Stat4 is expressed in neurons and immune cells, including activated neutrophils and macrophages, and the data presented here indicates an increased presence of activated Stat4 in cells in the hippocampal region in mice fed the HFD-C diabetogenic diet. In animals with LysM$^{cre}$-targeted deletion of Stat4, we found clear evidence of reduced activation of Stat4 in the hippocampus, and global Stat4 knockout animals exhibit marked improvement in the metabolic phenotype induced by the HFD-C diet in chow fed mice[16].

Numerous inflammatory cytokines in the brain are released in diabetes and neurodegenerative diseases, and it remains to be determined which inflammatory signals are essential for the phosphorylation and activation of Stat4 resulting in down-regulation of synaptic plasticity. IL-12 and IL-17 are leading candidates, given previous work showing that IL-17 can affect synaptic plasticity and microglial activation in a model of multiple sclerosis[44], IL-12 is involved in AD pathology and associated cognitive deficits[10–13], microglial IL-12 is upregulated in AD-like transgenic mice[14], and targeting the IL-12p40 subunit can reduce amyloid pathology and cognitive deficits in these animals[14]. However, the rescue of the effects of the HFD-C diabetogenic diet afforded by reduced Stat4 expression strongly suggests that this transcription factor could be a final common pathway linking cytokine-mediated inflammation to neurodegeneration in AD and other neuroinflammatory diseases.

$LysM^{cre}$ mediated deletion of Stat4 represents a targeted reduction of Stat4 in immune cells. However, more recent data looking at this promoter in both the brain (including hippocampus) and retina clearly indicates that a subset of neurons and microglia express the LysM$^{Cre}$ promoter[29,30,45]. Therefore, studies will be needed to further clarify if $Stat4^{\Delta LysM}Ldlr^{-/-}$ mice experience additional loss of activity of Stat4 in neurons and/or glia in the CNS. It is of great interest that recent studies have shown that 25-hydroxycholesterol can act via microglial activation to impair synaptic plasticity[46], and that restoring metabolism of myeloid cells can reverse cognitive decline due to aging[43].

A study of a natural herb extract of cinnamon bark found a mixture of inhibitory actions on p38, JNK, ERK1/2 and Stat4, but not Stat6, that modulated IFN-gamma expression and cytokine secretion in activated T cells[47]. It remains to be determined what effect selectively targeting Stat4 in myeloid cells, neurons or glia may have on declines in long-term, activity-dependent synaptic plasticity and cognition in normal aging or neurodegenerative diseases. However, our data suggests that suppression of inflammation by targeting Stat4, in both immune cells and neurons, may be a method for preventing, or even reversing, the loss of LTP observed in AD and other neurodegenerative diseases associated with impairments in cognitive function.

The data presented here support a role for the inflammatory transcription factor Stat4 in neuroinflammatory synaptotoxicity and cognitive impairments associated with HFD-C diets, atherosclerosis, and metabolic dysfunction. Specifically, LysM$^{cre}$-dependent deletion of Stat4 was completely protective of both impairments in long-term synaptic plasticity and metabolic changes elicited by sixteen weeks feeding with a HFD-C diet. Stat4 appeared to prevent aging-associated declines in magnitude of long-term potentiation (LTP) of synaptic transmission, suggesting that it plays an important role in inflammation that contributes to metabolic disease and aging and diabetes-associated cognitive decline. These data implicate Stat4 in inflammatory pathways associated with metabolic syndrome, atherosclerosis, neuroinflammation, and potentially AD, suggesting that treatments that suppress activity of this transcription factor or downstream pathway components could reduce AD risk in diabetics, and in normal aging.

## Methods

**Animals.** $Stat4^{\Delta LysM}Ldlr^{-/-}$ were generated via cross of newly-developed $Stat4^{fl/fl}$ mice[48] with $LysM^{Cre}$ (*JAX Labs, 6.129P2-*

*Lyz2tm1(cre)Ifo/J*) and *Ldlr*$^{-/-}$ (JAX Labs, B6.129S7-Ldlr$^{tm1Her}$/J) mice in our laboratories[3,4]. Male and female mice were fed a diabetogenic high fat/high cholesterol diet (HFD-C, BioServ No. F4997) or chow diet (Fisher LabDiet No. 0001319) for 16 weeks, from 8–24 weeks of age. Mice were used for brain slice preparation at either 3 or 24–26 weeks old, as indicated. All experiments were conducted under protocols approved by the NYMC Institutional Animal Care and Use Committee, in compliance with all relevant ethical regulations for animal testing.

**Fasting blood glucose and insulin tolerance test**. To perform glucose tolerance tests (GTT), 16 week HFD-C fed control *Stat4$^{fl/}$$^{fl}$Ldlr*$^{-/-}$ and *Stat4$^{fl/fl}$LysM$^{cre}$Ldlr*$^{-/-}$ mice ($n = 8$; male and female respectively of each genotype) were fasted overnight and then injected intraperitoneally with filter-sterilized 2 g/kg glucose in 0.9% NaCl as described previously[49,50]. A tail vein blood sample was taken before the injection and at 10, 20, 30, 60, 90, and 120 min time points post-injection for temporal measurements of blood glucose. Insulin tolerance tests (ITT) were conducted by intraperitoneally injection with insulin (0.75 U/kg) in 0.9% NaCl. Blood samples were taken immediately before and at 15, 30, 45, and 60 min post-injection.

**Immunofluorescence staining**. Immunofluorescence staining for phosphorylated Stat4 (pStat4) was performed on paraffin-embedded, 5-micron coronal mouse brain sections from 24-week-old and 3-week-old mice ($n = 4$ per group). Sections were baked at 60 °C for 1 h followed by the following deparaffinization protocol: slides were placed in 100% xylene three times for four minutes each, followed by 100% ethanol three times; than 95% ethanol two times; than one time in 70% ethanol for two minutes each. Slides were then placed in distilled water twice on a shaker. Heat-induced antigen retrieval was performed by microwaving slides in antigen retrieval solution (10 mM citrate, 0.05% Tween 20, pH 6.0). After cooling to room temperature, slides were rinsed in dH$_2$O and blocked with 2% donkey serum for 30 minutes. The slides were incubated with a mouse monoclonal IgG antibody recognizing pStat4 (Santa-Cruz 28296; 1:100 dilution) overnight at 4 °C. The following day, slides were washed three times with PBS for five minutes each. The secondary antibody Alexa Fluor 594 goat anti-mouse IgG (catalog number A11005, 1:500 dilution) was added to slides for two hours at room temperature. Slides were counterstained with DAPI and imaged using a Zeiss LSM 980 plus Airyscan II confocal laser microscope. Analysis was performed using NIH ImageJ of at least 3-4 sections per condition. Sections stained with secondary antibody only were used as negative controls (Supplementary Fig. 1).

**Nanostring expression profiling**. Four 24-week-old male mice from both genotypes, fed either the HFD-C from 8–24 weeks-of age, or chow diet for all 24 weeks, were deeply anesthetized under isoflurane, decapitated, and their brains quickly harvested and fixed using 10% formalin. After fixation, using two short micro spatulas, each hippocampus was manually dissected free from one hemisphere of the brain of each animal, placed back in 10% formalin and delivered to core services at the Albert Einstein College of Medicine. RNA was extracted from the hippocampus using Qubit and the concentration of each sample was measured. mRNA levels were determined using the NanoString (www.nanostring.com) platform utilizing a Host Response Panel, a custom Code Set which profiled 785 genes in mouse across 50+ pathways. One hundred nanograms of each total RNA sample was prepared as per the manufacturer's instructions. RNA expression was quantified on the nCounter Digital Analyzer and raw and normalized counts were generated with Rosalind

application (https://app.rosalind.bio). The WikiPathways platform was used to identify pathways altered between the analyzed groups of mice.

**Hippocampus slice electrophysiology**. Electrophysiological field potential recordings from Schaffer collateral-CA1 synapses in in vitro hippocampal slices were performed using standard methods as reported previously[51]. Mice were decapitated under deep isoflurane anesthesia, the brains quickly removed, hemisected, and cut with a vibratome (Leica model VT1200S) at a thickness of 350 μm. The tissue block was glued with cyanoacrylate adhesive to a stage immersed in ice-cold cutting solution composed of (in mM): 92 NMDG, 0.5 CaCl$_2$, 10 MgSO$_4$, 2.5 KCl, 1.25 NaH$_2$PO$_4$, 30 NaHCO$_3$, 20 HEPES, 25 Glucose, 2 Thiourea, 5 Sodium Ascorbate, 3 Sodium Pyruvate, 5 N-acetyl-L-cysteine oxygenated with 95% O$_2$/5% CO$_2$. After slicing, the hippocampal slices were transferred into a holding chamber containing cutting solution at room temperature for 5-10 minutes before transferred into another holding chamber, composed of (in mM): 92 NaCl, 2 CaCl$_2$, 2 MgCl$_2$, 2.5 KCl, 1.25 NaH$_2$PO$_4$, 30 NaHCO$_3$, 20 HEPES, 25 Glucose, 2 Thiourea, 5 Sodium Ascorbate, 3 Sodium Pyruvate, 5 N-acetyl-L-cysteine that bubbled with 95% O$_2$/5% CO$_2$. All slices were incubated at 30 °C for 45–50 min in the holding chamber continuously oxygenated with 95% O$_2$/5% CO$_2$, then moved to 32 °C for electrophysiological recording.

Once transferred to the recording chamber, slices were continuously perfused with bubbled (95% O$_2$/5% CO$_2$) aCSF which was composed of (in mM): 120 NaCl, 26 NaHCO$_3$, 2.5 KCl, 1.25 NaH$_2$PO$_4$, 2.5 CaCl$_2$, 1.5 MgCl$_2$, 10 Glucose and maintained at 32 °C. Borosilicate-glass recording electrodes (1–2 MΩ when filled with aCSF) were pulled with a Sutter micropipette puller (Model P-97, Sutter Instrument, Novato, CA), and inserted in the *stratum radiatum* of hippocampal field CA1 to record field excitatory post-synaptic potentials (fEPSPs). To elicit evoked responses, current pulses applied with stimulus intensity adjusted to evoke ~50% of maximal fEPSPs (50 pA to 100 pA; 100 μs duration) at 30 s intervals were delivered using a bipolar tungsten stimulating electrode that was placed in the Schaffer collateral/commissural fibers. The high-frequency theta burst stimulus (TBS) paradigm for induction of LTP consisted of 3 theta burst trains separated by 3 min, 10 bursts each, 5 pulses per burst, a burst frequency of 100 Hz and inter-burst interval of 200 ms. Electrical stimulation was delivered by an ISO-Flex isolator controlled by a Master 8 pulse generator (AMPI, Jerusalem, Israel) to evoke fEPSPs. Electrophysiological signals were amplified with a AC differential amplifier (Model 1700, A-M Systems, Sequim, WA, USA) and digitized via a DAQ device (MC Measurement Computing, Norton, MA), controlled and analyzed with SciWorks software (A-M Systems, Sequim, WA, USA) software run on a PC. The slopes of fEPSP were measured by linear interpolation from 20–80% of maximum negative deflection. Raw values of fEPSP at half-maximal stimulus intensity (SI) during 15 min baseline recording was averaged to compare baseline excitability across groups. The magnitude of LTP was measured and compared between *Stat4$^{fl/}$$^{fl}$Ldlr*$^{-/-}$ control mice ($N = 14$), and *Ldlr*$^{-/-}$ mice without the flox/flox insertion ($N = 26$). As shown in Supplementary Fig. 2A, LTP was not significantly different between the two groups of mice, indicating that the flox/flox insertion alone did not alter the magnitude of LTP (Student's *t*-test, $p = 0.33$). We also analyzed the magnitude of LTP separately in slices from *Stat4$^{fl/}$$^{fl}$Ldlr*$^{-/-}$ control and *Ldlr*$^{-/-}$ background mice, and found no differences between any of the groups (Supplementary Fig. 2B, $F(3,36) = 0.74$, $p = 0.54$). Finally, separate analysis of male and female *Stat4$^{fl/fl}$Ldlr*$^{-/-}$ control slices showed the significant effect of HFD-C diet on LTP, but no significant differences in the magnitude of LTP within each diet condition (Supplementary Fig. 3, Two-way

ANOVA, F(3,34) = 8.025, $p = 0.004$), so we pooled data from males and females for remaining experimental analyses.

For analysis of paired-pulse inhibition/facilitation of population spikes in CA1 neurons, the stimulating electrode was placed in the *stratum radiatum* layer to stimulate Schaffer collaterals, and the recording electrode in the CA1 *stratum pyramidale* layer. Population spike (PS) magnitudes were measured as the amplitude of the negative spike, extrapolated by drawing a tangent between the peak of the EPSP and the peak of the PS, and then taking the vertical distance from the negative peak of the PS to the tangent line. Population spikes were evoked at a stimulus intensity that elicited 50% of maximal amplitude in the first pulse spike. A series of inter-pulse intervals (IPIs), 20–500 ms in duration, was used to study synaptic properties of facilitation or depression. Four sweeps of population spikes were recorded for each IPI per slice, and the responses averaged. The ratio of the second evoked population spike amplitude (PS2) to the first (PS1) was used to determine depression or facilitation, with a ratio >1 corresponding to facilitation, and a ratio <1 indicating depression.

**Statistics and reproducibility**. All measurements for each n were taken from distinct brain slices or brain sections. Sample number per group was selected using power analyses with significance level preset to $P < 0.05$, to detect between group differences in magnitude of long-term potentiation, paired-pulse facilitation, and population excitatory postsynaptic potential slopes of 8-10% at a power of 85% with typical parameter standard deviations. Data were analyzed with Graphpad Prism v9 (GraphPad Software, Boston, MA, USA) or IBM SPSS v22 (IBM, Armonk, NY, USA) for One-way ANOVA for repeated measures. Comparisons between groups were made using Student's two-tailed t-test for unpaired data, with data expressed as mean ± SEM.

**Reporting summary**. Further information on research design is available in the Nature Portfolio Reporting Summary linked to this article.

## Data availability
Nanostring microarray data were deposited into the Gene Expression Omnibus database under accession number GSE240045. The experimental data that support the findings of this study are available in Figshare[52] at https://figshare.com/s/059d93a1209dd23951f0.

## Code availability
All computer code for data acquisition and analysis is commercially available.

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

## Acknowledgements

This work was supported by NIH R01HL142129 and HL142129-04S1 (EVG and JLN) and NIH DK R01105588 (JLN). We would like to acknowledge the Albert Einstein College of Medicine Diabetes Center Molecular Core Facility for assistance with Nanostring™ gene expression data analysis.

## Author contributions

X.Z., C.H., M.K., M.D., H.M., and X.Y. conducted the experiments; X.Z., C.H., M.K., R.B., C.K., E.G., and P.S. analyzed data; E.G., J.N. and P.S. prepared the manuscript; J.N. and P.S. conceived and supervised the project.

## Competing interests

The authors declare the following competing interests: U.S. provisional patent: "USE OF STAT4 INHIBITORS FOR PREVENTION AND TREATMENT OF ALZHEIMER'S DISEASE". Patent Applicant: New York Medical College. Name of Inventors: Jerry L. Nadler, Patric K. Stanton. Application Number: 20230233595, Status: Published, Aspect of manuscript covered in the provisional patent: The observations in this manuscript, that deficiency in Stat4 protects long-term synaptic plasticity from the effects of a high-fat, high-cholesterol Western diet, support the hypothesis of this provisional patent that inhibitors of Stat4 have the potential to slow or prevent the development of Alzheimer's disease, which is also promoted by a Western diet, and also involves activation of neuroinflammatory pathways.
