## [Peer Review File · Communications Biology]

Reviewers' comments:

Reviewer #1 (Remarks to the Author):

The authors "hypothesize that STAT4 is one final common pathway mediator of inflammatory damage that is important in normal aging, diabetes and Alzheimer's disease".

The authors generated Stat4^{-/-} LdLr^{-/-} mice and Stat4^{fl/fl} LySMCre LdLr^{-/-} mice, which lack STAT4 under control of the LysMCre promoter. These mice were fed with high fat/high cholesterol (HFD-C) or chow diets for 16 weeks.

In slices from LdLr^{-/-} mice fed with chow long-term potentiation (LTP) was moderate, whereas in slices from Stat4^{f/f} LdLr^{-/-} mice fed with HFD-C LTP was dampened. In slices from Stat4^{f/f} LySMCre LdLr^{-/-} mice, fed with HFD-C, LTP was bigger than that from Stat4^{f/f} LdLr^{-/-} (Fig.4).

Major concerns:

1) In the introduction, they state that: (1) Nothing is currently available to slow or stop the course of Alzheimer's disease (AD), (2) IL-12 is involved in AD and IL-12 activated STAT4, (3) There is a link between AD and metabolic syndrome and LTP is impaired in AD models and aging. (4) they tested the hypothesis that STAT4 plays a key role in synaptic plasticity. There is a gap between this introduction and the current results. I do not think that AD is the main stream of the current work. The authors should state the general nature of STAT proteins. In this work, the important effort is the establishment of STAT-deficient LdLr^{-/-} and Stat4^{fr/fr} LySMCre LdLr^{-/-} strains. The authors should state the characters of these two strains. First of all, the nature of LdLr^{-/-} mice should be explained. Without information about LdLr^{-/-}, it is difficult to follow. The introduction should be rewritten.

2) Their conclusion is not clear in spite of the long conclusion paragraph. In the abstract they suggest that STAT4 activation reduces aging-associated inflammation and risk of Alzheimer's disease; AD and aging have little to do with the current results. Since all of their results are from modified strains the authors should be careful in presenting a conclusion but we need a concise conclusion.

3) The authors describe Stat4^{fl/fl} LdLr^{-/-} as "control" at several places. But I do not think that Stat4^{fl/fl} LdLr^{-/-} mice can serve as control in this work. LdLr^{-/-} mice in the same condition, I mean fed with high fat, should be the control. If the authors do not provide data from LdLr^{-/-} fed with HFD-C, can they provide any rationales for not showing the real control? This is the top concern.

4) LTP is easily affected by estrus cycle. Some LTP investigators avoid diestrus. As supplemental data, re-analysis of Fig. 4 data after separation of males and female may provide better results.

Minor points

page 2

5) In the abstract, "these findings suggest that suppressing STAT4 activation can reduce aging-associated inflammation and protect against effects of high-fat/high cholesterol diets on cognition, type 2 diabetes, reducing risk of Alzheimer's disease and other neurodegenerative disorders". This description is too much from the current findings.

page 3-4

6) Do they need abbreviation CVD for cardiovascular disease? Also, do they need abbreviation CTE for chronic traumatic encephalopathy?

page 4

7) Cholesterol levels should be monitored after a certain period of feeding.

8) Similarly, glucose levels (or urine glucose) should be monitored if the author believe that the development of diabetes plays a pivotal role in cognitive dysfunction.

page 5

9) What does p stand for in "pSTAT4"?

page 8-9 RESULTS

10) "Figure 1 illustrates....., compared with control Stat4fl/fl Ldlr-/- mice." "At three weeks of age,....compared to age-matched control mice with normal STAT4 expression. Do the author want readers to sense subtle nuance between "with" and "to"?

11) In the first paragraph of RESULTS, the authors describe "This confirms that the high-fat/high-cholesterol diet activates STAT4". Without data from mice fed with chow, it is difficult to determine so.

page 11 DISCUSSION

12) In the second paragraph of DISCUSSION, the second sentence is long and not easy to follow. "First,....."

page 13

13) DISCUSSION reads "targeting STAT4 could be a new therapeutic approach". It has been reported that cinnamon inhibits STAT4 (Beom-Joon Lee et al., 2011. doi: 10.3109/08923973.2011.564185). The authors may want to discuss about cinnamon's actions.

14) It has been reported that high fat diet induces cognitive impairment (Gainey SJ et al., 2016 DOI: 10.3389/fnbeh.2016.00156, Duffy C Met et al., 2016 DOI: 10.1016/j.nlm.2018.11.008, Spencer S et al., 2017 DOI: 10.1016/j.neurobiolaging.2017.06.014). Some of these papers should be cited.

15) Figure 4 legend. STAT5 should be STAT4.

16) In FIGURE LEGENDS and figures the author use LDLr-/-, but in the text Ldlr-/- and sometimes LDRr-/-.

Reviewer #2 (Remarks to the Author):

In this work Zhang et al. show that the activation of the transcription factor STAT4 participates in the reduction of LTP occurring in type 2 diabetes mellitus (T2DM) paradigm. This result is interesting and potentially relevant, but the work as a whole is in a preliminary phase. Below I give a series of suggestions that in my opinion would make this a more solid story, relevant to the field of study of the mechanisms involved in the association between type 2 diabetes and Alzheimer's disease and, perhaps, in the development of agents that can interfere with the progression from T2DM to pathological cognition.

Figure 1 is very uninformative, of little value. The authors should provide the following data: 1) that metabolic manipulation produces the change in Stat4 activity in the brain of control mice and 2) that the model mice are a valid tool for the reduction of Sta4 in the brain, by other methods besides immunofluorescence. Demonstrating reduced activity by immunofluorescence is very unconvincing, even if an antibody against the phosphorylated form is used. The image shown gives no indication of what brain area one is looking at. I request to show a lower magnification of brain distribution of pSTAT4 so to be able to compare levels in different brain regions, with special focus on the hippocampus. In addition, authors should show data from biochemical or molecular readouts, such as qPCR of target or upstream genes, cytokine assays or chromatin occupancy. The authors must consider that the main genetic association of STAT4 with diabetes is with type 1 diabetes (early onset, not even late onset Type1 diabetes), so it is important that they show data on STAT4 activity in the brain of mice in control and type 2 diabetes state, with normal and reduced (KO) STAT4.

Fig. 2. The authors need to show the effect of diet on brain insulin signaling (activity of insulin receptor and substrates), not just that STAT4 KO have better systemic metabolic parameters when

subjected to HFD-C conditions: i) do mice with physiological levels of STAT4 show signs of hippocampal insulin resistance when exposed to HFD-C?, ii) does STAT4 KO prevent the occurrence of hippocampal insulin resistance in the HFD-C mice?, iii) what synaptic plasticity (biochemical) pathways is rescued when this TF is knocked out, CamKII/PKA, Ras-Erk, Ras-PI3K? Some type of information on these lines is important to have more solid knowledge of how Stat4 could be contributing to the worsening of cognitive aspects in individuals with T2DM; i.e. to what extent there is a central alteration mediated by the lack of STAT4 -insulin resistance? - or is it all consequences of the lack of STAT4 at peripheral level with central (hippocampal) consequences. And regardless of the answer: how does activation of STAT4 in a diabetes-like condition lead to reduced LTP?

Fig. 3. I do not see too much relevance for this work to show as a final figure that STA4 does not have any role in LTP in 3-week-old mice, even more so when the mice have not been subjected to any type of alteration that could induce the activation of STAT4. These are data that could easily fit as supplementary information.

Fig. 4. The authors go a long way in saying that their results support the hypothesis that STAT4 is a key mediator in the onset of AD in individuals with type 2 diabetes. I think they should postulate this possibility in the discussion and not in the description part of the electrophysiological result.

Zhang et al. COMMSBIO-23-0966-T - Response to the Reviewers

We are deeply grateful to both reviewers for their appreciative comments and constructive criticisms. We have addressed each of the concerns raised by the reviewers in the revised manuscript. The changes in the amended manuscript are as follows:

Reviewer #1:

1. *In the introduction, they state that: (1) Nothing is currently available to slow or stop the course of Alzheimer's disease (AD), (2) IL-12 is involved in AD and IL-12 activated STAT4, (3) There is a link between AD and metabolic syndrome and LTP is impaired in AD models and aging. (4) they tested the hypothesis that STAT4 plays a key role in synaptic plasticity. There is a gap between this introduction and the current results. I do not think that AD is the mainstream of the current work. The authors should state the general nature of STAT proteins. In this work, the important effort is the establishment of STAT-deficient Ldlr^{-/-} and Stat4^{fl/fl} LysM-Cre LDLr^{-/-} strains. The authors should state the characters of these two strains. First of all, the nature of LDLr^{-/-} mice should be explained. Without information about LDLr^{-/-}, it is difficult to follow. The introduction should be rewritten.*

2. *Their conclusion is not clear in spite of the long conclusion paragraph. In the abstract they suggest that STAT4 activation reduces aging-associated inflammation and risk of Alzheimer's disease; AD and aging have little to do with the current results. Since all of their results are from modified strains the authors should be careful in presenting a conclusion but we need a **concise conclusion**.*

We have rewritten both the Introduction and Discussion, to clarify characteristics of the strains we studied, and added results of Nanostring analysis of the differences between the Stat4^{fl/fl}Ldlr^{-/-} and Stat4^{ΔLysM}Ldlr^{-/-} mice in expression of genes, under conditions of both chow and high-fat, high-cholesterol (HFD-C) diet consumption (pp. 9-12, yellow highlighting; pp. 15-16, blue highlighting). While one observation of these new data is that the HFD-C diet increased expression of APP significantly less in Stat4^{fl/fl}Ldlr^{-/-} than in Stat4^{ΔLysM}Ldlr^{-/-} control mice, we have softened our speculation concerning the possible connection between STAT4 activation and Alzheimer's disease, and moved it only into the Discussion. We also added text clarifying our reasoning for using Ldlr^{-/-} mice as a mouse model of atherosclerosis and provided additional information on the general nature of STAT4 proteins, generation of Stat4^{ΔLysM}Ldlr^{-/-} and Stat4^{fl/fl}Ldlr^{-/-} mice.

3. *The authors describe Stat4^{fl/fl}LDLr^{-/-} as "control" at several places. But I do not think that Stat4^{fl/fl}LDLr^{-/-} mice can serve as control in this work. LDLr^{-/-} mice in the same condition, I mean fed with high fat, should be the control. If the authors do not provide data from LDLr^{-/-} fed with HFD-C, can they provide any rationales for not showing the real control? **This is the top concern.***

Editor: Actually, a very fast response from Reviewer #1. We both agree that getting floxed animals on a control diet would be a good comparison (particularly in Fig. 4), though I would also lean toward including LysM-Cre/Ldlr^{-/-} animals (to rule out artifacts from Cre recombinase expression).

We performed additional experiments on the requested control group, the Stat4^{fl/fl}Ldlr^{-/-} (new Figs. 6,7,8,9 and S2). LTP was significantly reduced by 16 weeks of the high-fat, high-cholesterol diet in the Stat4^{fl/fl}Ldlr^{-/-} control mice, but was not impaired in the Stat4^{ΔLysM}Ldlr^{-/-} mice. We have added these data to the Results.

We generated a new Stat4^{fl/fl}Ldlr^{-/-} mice from a cross of Ldlr^{-/-} mice with a recently developed in our laboratory Stat4^{fl/fl} mice. As the floxed Stat4^{fl/fl} mice is a relatively new strain, we decided to use floxed Stat4^{fl/fl}Ldlr^{-/-} mice as a control for Stat4^{ΔLysM}Ldlr^{-/-} mice in order to account for a

potential effect of *Stat4^{fl/fl}* in our experiments. We clarified this on pg. 8, lines 10-12 of the revised manuscript.

We also performed additional experiments to investigate how high fat diet feeding (HFD-C) modulates LTP in *Stat4^{fl/fl}Ldlr^{-/-}* mice. Chow fed *Stat4^{fl/fl}Ldlr^{-/-}* mice at 24 weeks of age (Fig.5) exhibited robust LTP of 1.76 ± 0.14 times pre-TBS baseline responses at 60 minutes post-TBS. When age-matched *Stat4^{fl/fl}Ldlr^{-/-}* mice were fed the HFD-C diet for 16 wks, the magnitude of LTP was significantly reduced (1.25 ± 0.04 times pre-TBS baseline, ***, $P=0.001$, $t=4.286$, $df=36$, Student's t-test), indicating that the HFD-C diet impairs the long-term, activity-dependent synaptic plasticity necessary for normal cognition, learning and memory. We have added these results to the Results section (pg. 12-14).

4) *LTP is easily affected by estrus cycle. Some LTP investigators avoid diestrus. As supplemental data, re-analysis of Fig. 4 data after separation of males and female may provide better results.*

We have re-analyzed the data from Fig. 4 separating males and females, which showed no significant differences in effects of lack of STAT4 on LTP, and added this as Supplemental Fig. 3.

Minor points

5) *In the abstract, "these finding suggest that suppressing STAT4 activation can reduce aging-associated inflammation and protect against effects of high-fat/high cholesterol diets on cognition, type 2 diabetes, reducing risk of Alzheimer's disease and other neurodegenerative disorders". This description is too much from the current findings.*

We have rewritten this sentence as follows:

"Nanostring analysis revealed a number of neuroinflammatory and synaptic plasticity genes that were reduced in expression by the HFD-C diet, and which were less reduced in *Stat4^{ΔLysM}Ldlr^{-/-}* mice challenged with the HFD-C diet. Taken together, our data suggest that suppression of STAT4 activation may be a novel approach for protecting against effects of western high-fat/high cholesterol diets on cognition, type 2 diabetes, and may reducing risk of Alzheimer's disease and other neurodegenerative disorders associated with neuroinflammation."

6) *Do they need abbreviation CVD for cardiovascular disease? Also, do they need abbreviation CTE for chronic traumatic encephalopathy?*

You are correct, we have removed the abbreviations.

7) *Cholesterol levels should be monitored after a certain period of feeding.*

It is known that HFD-C diets in *Stat4^{fl/fl}Ldlr^{-/-}* mice leads to increased total circulating cholesterol (~ 800 md/dl) and that LysMCre promoter does not modify these changes. We have added a citation from our previous study (Keeter et al, *Frontiers in Cardiovascular Medicine*, 2023), and added this sentence to the Methods on pg. 4:

"Keeter et al, Neutrophil-specific STAT4 deficiency attenuates atherosclerotic burden and improves plaque stability via reduction in neutrophil activation and recruitment into aortas of *Ldlr^{-/-}* mice. *Frontiers in Cardiovascular Medicine*, 2023 (in press)"

8) *Similarly, glucose levels (or urine glucose) should be monitored if the author believe that the development of diabetes plays a pivotal role in cognitive dysfunction.*

In the animals we studied here, both glucose and insulin tolerance tests show directly that the HFD-C diet results in metabolic dysfunction, and that this was associated with alterations in synaptic plasticity. We did not mean to state that these animals have diabetes, and have ensured that this data suggests, but does not yet prove, that develop of diabetes *per se* plays a pivotal role in plasticity and cognitive dysfunction.

9) *What does p stand for in “pSTAT4”?*

Phosphorlyated STAT4, we have clarified this in the text.

page 8-9

RESULTS

10) *“Figure 1 illustrates....., compared with control Stat4fl/fl Ldlr-/- mice.” “At three weeks of age,....compared to age-matched control mice with normal STAT4 expression. Do the authors want readers to sense subtle nuance between “with” and “to”?*

Sorry, we didn’t imply a subtle nuance, and have changed to “with” throughout the sentence.

11) *In the first paragraph of RESULTS, the authors describe “This confirms that the high-fat/high-cholesterol diet activates STAT4”. Without data from mice fed with chow, it is difficult to determine so.*

We have rewritten this sentence in the Results to state more precisely that “This confirms that cells in this region of the hippocampus express activated STAT4, and that there are residual cell types still expressing STAT4 in the Stat4^{ΔlysM}Ldlr^{-/-} mice.”

page 11 DISCUSSION

12) *In the second paragraph of DISCUSSION, the second sentence is long and not easy to follow. “First,.....”*

We have shortened this sentence as follows:

“The HFD-C induced impairments in long-term plasticity we observed in adult, but not three week old, animals is consistent with reports that such diets can alter the developmental time course of activity-dependent synaptic plasticity in the non-diseased brain.”^{36,37} “

13) *DISCUSSION reads “targeting STAT4 could be a new therapeutic approach”. It has been reported that cinnamon inhibits STAT4 (Beom-Joon Lee et al., 2011. doi: 10.3109/08923973.2011.564185). The authors may want to discuss about cinnamon’s actions.*

We thank the reviewer for alerting us to this interesting and relevant study, which we have added to the Discussion on pg. 17.

14) *It has been reported that high fat diet induces cognitive impairment (Gainey SJ et l., 2016 DOI: 10.3389/fnbeh.2016.00156, Duffy C Met al., 2016 DOI: 10.1016/j.nlm.2018.11.008, Spencer S et al., 2017 DOI: 10.1016/j.neurobiolaging.2017.06.014). Some of these papers should be cited.*

We have added citations of these papers in the Discussion pg. 15.

15) Figure 4 legend. STAT5 should be STAT4.

Corrected

16) In FIGURE LEGENDS and figures the author use LDLr^{-/-}, but in text Ldlr^{-/-} and sometimes LDRr^{-/-}

Corrected

Reviewer #2:

In this work Zhang et al. show that the activation of the transcription factor STAT4 participates in the reduction of LTP occurring in type 2 diabetes mellitus (T2DM) paradigm. This result is interesting and potentially relevant, but the work as a whole is in a preliminary phase. Below I give a series of suggestions that in my opinion would make this a more solid story, relevant to the field of study of the mechanisms involved in the association between type 2 diabetes and Alzheimer's disease and, perhaps, in the development of agents that can interfere with the progression from T2DM to pathological cognition.

1) *Figure 1 is very uninformative, of little value. The authors should provide the following data: 1) that metabolic manipulation produces the change in Stat4 activity in the brain of control mice and 2) that the model mice are a valid tool for the reduction of Sta4 in the brain, by other methods besides immunofluorescence. Demonstrating reduced activity by immunofluorescence is very unconvincing, even if an antibody against the phosphorylated form is used. The image shown gives no indication of what brain area one is looking at. I request to show a lower magnification of brain distribution of pSTAT4 so to be able to compare levels in different brain regions, with special focus on the hippocampus. In addition, authors should show data from biochemical or molecular readouts, such as qPCR of target or upstream genes, cytokine assays or chromatin occupancy. The authors must consider that the main genetic association of STAT4 with diabetes is with type 1 diabetes (early onset, not even late onset Type1 diabetes), so it is important that they show data on STAT4 activity in the brain of mice in control and type 2 diabetes state, with normal and reduced (KO) STAT4.*

We performed Nanostring analysis of the expression of a panel of inflammatory and nervous system genes, that show 1) extensive alterations in brains of mice lacking STAT4 under LysM^{cre} promoter when fed a normal chow diet, and 2) protection against the effects of HFD-C diet on gene expression patterns, comparing Stat4^{ALysM}Ldlr^{-/-} mice and Ldlr^{-/-} mice control mice. These data are including in new Results sections highlighted in the manuscript.

We have amended Fig. 1 as requested to supply lower magnification fluorescence images and companion hematoxylin/eosin stained sections illustrating the hippocampal region imaged.

2) *Fig. 2. The authors need to show the effect of diet on brain insulin signaling (activity of insulin receptor and substrates), not just that STAT4 KO have better systemic metabolic parameters when subjected to HFD-C conditions: i) do mice with physiological levels of STAT4 show signs of hippocampal insulin resistance when exposed to HFD-C?, ii) does STAT4 KO prevent the occurrence of hippocampal insulin resistance in the HFD-C mice?, iii) what synaptic plasticity (biochemical) pathways is rescued when this TF is knocked out, CamKII/PKA, Ras-Erk, Ras-PI3K? Some type of information on these lines is important to have more solid knowledge of how Stat4 could be contributing to the worsening of cognitive aspects in individuals with T2DM; i.e. to what extent there is a central alteration mediated by the lack of STAT4 -insulin resistance? - or is it all consequences of the lack of STAT4 at peripheral level with central (hippocampal) consequences. And regardless of the answer: how does activation of STAT4 in a diabetes-like condition lead to reduced LTP?*

In this paper, we found that STAT4 in cells expressing LysM plays a role in mediating glucose intolerance and insulin resistance. The deletion of STAT4 using the LysM^{cre} promoter improved peripheral glucose tolerance and sensitivity to insulin action. It will be important in future to perform a detailed analysis of changes in brain insulin resistance, but this was not the focus of the current study. However, our observation that thioredoxin-interacting protein (TXNIP) expression was reduced in brain of *Stat4^{ΔLysM}Ldlr^{-/-}* mice on the HFD-C diet suggests there may be metabolic improvement in these mouse brains. As mentioned in the Results on pg. 11, TXNIP-deficient mice have been shown to be protected from diet-induced insulin resistance.

3) *Fig. 3. I do not see too much relevance for this work to show as a final figure that STA4 does not have any role in LTP in 3-week-old mice, even more so when the mice have not been subjected to any type of alteration that could induce the activation of STAT4. These are data that could easily fit as supplementary information.*

We wish to respectfully suggest that it is important for the interpretation of our findings in 24 week old mice that we have controlled for possible effects of the lack of STAT4 on early development of the brain and synaptic function. We have emphasized that this is the rational for performing these the same measurements at 3 and 24 weeks of age.

4) *Fig. 4. The authors go a long way in saying that their results support the hypothesis that STAT4 is a key mediator in the onset of AD in individuals with type 2 diabetes. I think they should postulate this possibility in the discussion and not in the description part of the electrophysiological result.*

We thank both reviewers for highlighting places where are conclusions were too strong for the current evidence. We have moved our consideration of possible interpretations to the Discussion.

REVIEWERS' COMMENTS:

Reviewer #1 (Remarks to the Author):

The authors made substantial additions in data, especially in the control. Descriptions were also improved. I don't have further suggestions.

Reviewer #2 (Remarks to the Author):

The authors have added sufficient new experimental data, reanalyzed and graphed others, and responded respectfully and correctly to my comments. The paper can be accepted for publication.